

# Conflict of interest explains the size of student evaluation of teaching and learning correlations in multisection studies: a meta-analysis

Bob Uttl[1], Kelsey Cnudde[2] and Carmela A. White[3]

[1] Mount Royal University, Calgary, AB, Canada
[2] University of Calgary, Calgary, AB, Canada
[3] University of British Columbia Okanagan, Kelowna, BC, Canada

Corresponding author
Bob Uttl, uttlbob@gmail.com

## ABSTRACT

We examined the associations between the size of student evaluation of teaching and learning (SET/learning) correlations and presence of several conflicts of interest (COIs) including corporate, administrative, evaluation unit, SET author, and funder interests. Our meta-analyses of SET/learning correlations reported by multisection studies show that researchers with a vested interest in finding large positive SET/learning correlations found, on average, large positive SET/learning correlations. In contrast, researchers with no identifiable COIs found that SET/learning correlations were zero or nearly zero. The largest SET/learning correlations were reported by authors with ties to SET selling corporations. Smaller but still substantial SET/learning correlations were reported by researchers with administrative assignments and by researchers in evaluation units/departments responsible for the administration of SET. Moreover, authors with the most significant COIs were publishing their studies primarily prior to 1981 whereas authors with no or less significant COIs were publishing their studies in 1981 or afterwards. Studies published prior to 1981 reported small but significant ($r = .31$) SET/learning correlations whereas studies published in 1981 and after reported near zero, non-significant SET/learning correlations ($r = .06$). The presence of COIs was associated with earlier publication date but also with smaller samples. Finally, whereas corporate, administrative, and evaluation unit authors nearly ceased publishing multisection studies on SET/learning correlations, authors from business and economics departments are now responsible for the substantial portion of newer, larger, and higher quality studies published in 1981 and after.

## INTRODUCTION

For decades, colleges and universities have been asking students to evaluate the teaching effectiveness of their professors using a variety of Student Evaluation of Teaching (SET) questionnaires based on a widespread belief that students learn more from more highly rated professors. The SET are typically administered during the last few weeks of classes,

before students receive their final grades, and they ask students to rate their professors' teaching, for example, their overall teaching effectiveness as well as their knowledge, clarity, organization, friendliness, fairness, approachability, availability, etc. The SET ratings for each course/class are summarized, often by calculating means for each rated item and across all items, and the mean SET ratings are then used as a measure of professors' teaching effectiveness. The SET ratings are subsequently used for a variety of high stakes personnel decisions such as faculty hiring, firing, promotion, tenure, merit pay, teaching awards, and distinguished professor awards (*Spooren, Brockx & Mortelmans, 2013*; *Stark & Freishtat, 2014*; *Uttl, White & Gonzalez, 2017*; *Wachtel, 1998*).

The main evidence cited for the belief that students learn more from professors with higher SET ratings are several meta-analyses of multisection studies showing small-to-moderate correlations between SET ratings and student learning (*Uttl, White & Gonzalez, 2017*). For example, *Cohen (1981)* claimed that the correlation between overall instructor SET ratings and learning was .43 to .44. Similarly, *Feldman (1989)* claimed that the correlations between various SET item ratings and learning were as high as .57. These meta-analyses were reviewed by numerous researchers who all concluded that these meta-analyses are strong evidence of the validity of SET ratings as a measure of student learning. For example, *Abrami & d'Apollonia (1999*, p. 519) wrote: "[SET] ratings explain instructor impact on student learning to a moderate extent (corrected $r = .47$)." *Marsh (2007*, p. 339) wrote: "Perhaps more than any other area of SET research, results based on the multisection validity paradigm support the validity of SET." *Benton & Cashin (2012*, p. 4) wrote: "the multisection studies show that classes in which the students gave the instructor higher ratings tended to be the ones where the students learned more." Not surprisingly, these summary reviews then made their way as established facts into various self-help books for new faculty members. For example, *Davis* (*2009*, p. 534) wrote: "Students of highly rated teachers achieve higher final exam scores, can better apply course material, and are more inclined to pursue the subject subsequently."

Recently, Uttl and colleagues (*2017*) published a detailed review of all previously published meta-analyses on SET/learning correlations and found that they all suffered from multiple critical methodological shortcomings. For example, none of these meta-analyses adequately considered the possibility that their results may be an artifact of a small sample bias even though the smallest studies (with 5 sections) showed correlations as high as .91, whereas the larger studies (with 30 or more sections) showed much smaller correlations. The substantial relationship between the sample size and the size of the SET/learning correlations would have been obvious to anyone who plotted the data (*Uttl, White & Gonzalez, 2017*). Moreover, the previous meta-analyses did not adequately describe the search for and identity of all primary studies, often did not provide effect sizes and sample sizes for each primary study, often failed to weigh SET/learning correlations by sample size, and in general, were so poorly described as to be unreplicable. In one remarkable review of previous meta-analyses, *Abrami, Cohen & d'Apollonia (1988)* were concerned about "disparate conclusions of the [multisection study] reviews" (p. 151), discussed "why the reviews differ" (p. 155), and even considered the possibility of a publication bias (one type of small-sample bias) but then dismissed it. They puzzled over "troublesome" (p. 160)

disagreement between SET/learning correlations extracted by *Cohen (1983)* and *McCallum (1984)*. However, instead of checking the extracted values against the primary studies and determining the source of the disagreement, Abrami et al. chose to speculate about possible sources of disagreement. As *Uttl, White & Gonzalez (2017)* found, the principal reason for the "troublesome" disagreement was *McCallum*'s (*1984*) failure to correctly extract relevant values from the primary studies.

When *Uttl, White & Gonzalez (2017)* re-analyzed the previously published meta-analyses of multisection studies, they found that the previous findings were artifacts of small sample biases. The small sample size studies showed large SET/learning correlations whereas the large sample size studies showed no or only minimal SET/learning correlations. When the small sample bias was taken into account through a variety of meta-analytic methods, the estimated SET/learning correlations were much smaller than previously reported. Moreover, given the shortcomings of the previous meta-analyses, Uttl et al. conducted a new up-to-date meta-analysis of the SET/learning correlations from multisection studies. Their search yielded 51 articles with 97 multisection studies. Using limit meta analysis to adjust for the prevalence of small sample bias, the new data set resulted in a SET/learning correlation of only $r = .12$, 95% CI [.03–.21]. Moreover, when the same analysis was re-run but only with data from studies with prior knowledge/ability adjustments, the limit meta-analysis resulted in a SET/learning correlation of $r = -.06$, 95% CI [−.17–.07]. Accordingly, Uttl et al. concluded that the data from multisection studies do not support the claims that students learn more from more highly rated professors. Rather, the results from multisection studies provide evidence that SET ratings and learning are unrelated.

Why is it that for three to four decades fatally flawed meta-analyses of multisection studies by *Cohen (1981)* and *Feldman (1989)* have been cited as evidence of SET/learning correlations, and that SET ratings are considered valid enough to be used for making high stakes personnel decisions, including for terminating people's careers? How is it possible that experts who reviewed these meta-analyses did not notice major red flags in previous meta-analyses of multisection studies (e.g., not reporting SET/learning correlations and sample sizes for primary studies, not weighing SET/learning correlations by sample size), or in the primary studies themselves (e.g., impossibly high correlations in small sized primary studies)? How is it possible that when experts noticed "troublesome" disagreements between the SET/learning correlations in different meta-analyses they did not follow up and identify the source of the disagreements and instead chose to speculate about them?

One possibility, which we investigated for this article, is that the primary findings and the reviews of SET/learning correlations have been distorted by conflicts of interest (COI). One of the widely accepted definitions states that "a conflict of interest is a set of circumstances that creates a risk that professional judgments or actions regarding a primary interest will be unduly influenced by a secondary interest" (*Institute of Medicine, Board on Health Sciences Policy, Committee on Conflict of Interest in Medical Research, Education, and Practice, 2009*, p. 46). Consistent with this definition, when using SET ratings for making high stakes personnel decisions, the 'primary interests' include the promotion and safeguarding of the integrity of research, the validity of scientific findings and conclusions, the welfare of those to whom the scientific findings are applied, and the welfare of the

society that relies on those scientific findings. In contrast, secondary interests include direct financial gains by individual researchers such as profits, dividends, salaries, and other financial benefits. Secondary interests are also non-financial benefits such as efficient time-saving evaluations and management of faculty by chairs and deans, approval and praise from upper levels of the university administration, greater potential for administrative advancement, and the appearance of objectivity and public accountability (*Uttl, White & Gonzalez, 2017*). Although the financial benefits are often easier to determine and quantify objectively, the non-financial benefits may not be any less damaging to the primary interests. Importantly, by the definition above, a COI exists independently of an individual actually being influenced by a secondary interest. A COI is a set of circumstances that increases the risk of undue influence (*Institute of Medicine, Board on Health Sciences Policy, Committee on Conflict of Interest in Medical Research, Education, and Practice, 2009*).

The risk that the primary research findings on SET and the written reviews were influenced by COIs seems substantial. First, even a superficial glance makes it apparent that many multisection studies were authored by researchers who work for or own corporations that sell SET systems. For example, John Centra was heavily involved in the development of the Student Instructional Report (SIR) and SIR-II SET system by the Educational Testing Service (*Centra, 1977*; *Centra, 2015*); Peter Frey developed the Endeavor Instructional Rating Card and founded Endeavor Information Systems, Inc. (*Endeavor Information Systems Inc., 1979*); Herbert Marsh developed the Student Evaluation of Educational Quality (SEEQ) SET system, and he founded and became President of Evaluation, Testing and Research, Inc. (*Marsh, Fleiner & Thomas, 1975*); and Lawrence Aleamoni developed the Aleamoni Course/Instructor Evaluation Questionnaire (CIEQ), and he founded and became President of Comprehensive Data Evaluation Services Inc. (CODES, Inc.) (*Armen, 2016*; *Hill, 2006*), and CODES Inc. presently sells CIEQ through http://www.cieq.com/. Second, the experts who reviewed the previous meta-analyses of multisection studies and found that the meta-analyses provide strong evidence of the validity of SET as a measure of teaching effectiveness had strong interests in that particular conclusion. For example, *Cohen*'s (*1981*) meta-analysis, the first and the most highly cited meta-analysis of multisection studies claiming that SET/learning correlations were greater than $r = .40$, was based on *Cohen*'s (*1980*) dissertation. P. A. Cohen's dissertation was co-supervised by James A. Kulik and Wilbert J. McKeachie, known proponents of SET ratings who concluded that SET/learning correlations were small to moderate in their published narrative reviews of the same literature prior to P. A. Cohen's work (*Cohen, 1981*). Moreover, Kulik was an Associate Director, Center for Research on Learning and Teaching, University of Michigan (*Kulik, Kulik & Cohen, 1980*); P. A. Cohen became Research Associate in the same center by Fall 1980 after completing his dissertation (*Kulik, Kulik & Cohen, 1980*); and by the time he published his meta-analysis in 1981 he was an Assistant Director of the Office of Instructional Services and Educational Research, Dartmouth College (*Cohen, 1981*). Benton worked for the IDEA Center, a nonprofit organization selling the IDEA SET to colleges and universities worldwide (*Benton & Cashin, 2012*). Marsh and Aleamoni, as noted above, both developed their own SET systems and founded their own companies to distribute them. Third, a number of authors of multisection studies worked for units
or departments responsible for the evaluation of faculty teaching effectiveness in various colleges and universities. Finally, some multisection studies were funded by grants from corporations with vested interests, for example, Endeavor Information Systems Inc.

Notably, COIs tying authors to their SET enterprises were often not declared and readers may not know about them. For example, we only found out that Aleamoni was the president of Comprehensive Data Evaluation Services Inc.—a corporation that publishes and distributes Aleamoni's CIEQ SET—by discovering the case *Aleamoni v. Commissioner of Internal Revenue Service* where this information is disclosed (*Armen, 2016*). He, his wife, and his children owned 100% of shares in this company.

Moreover, even though *Clayson*'s (*2009*) findings are uninterpretable and his conclusions unwarranted (*Uttl, White & Gonzalez, 2017*), *Clayson (2009)* suggested that the magnitude of SET/learning correlations may vary by publication year, authors' department, and other factors. Our own review of multisection studies suggests that, in addition to study size and a variety of COIs, SET/learning correlations may be larger for studies published prior to *Cohen*'s (*1981*) meta-analysis compared to those published after, as well as studies originating from education or psychology departments vs. studies originating from business and economics departments. In general, studies published prior to 1981 were conducted primarily by SET corporations and evaluation units, whereas studies published after 1981 were conducted primarily by authors from business and economics departments.

Accordingly, the current study had several main goals. The first goal was to determine whether SET/learning correlations reported in primary studies were larger for studies published prior to 1981 vs. 1981 and after. *Cohen*'s (*1981*) meta-analysis appeared to have cemented a belief that SET/learning correlations were substantial. However, as noted above, a majority of studies published prior to 1981 were published by authors with corporate, administrative, and evaluation unit COIs, and thus, to the extent to which COIs play a role, we would expect the earlier studies to report larger SET/learning correlations. In addition, earlier vs. later studies employed smaller samples, and thus, in combination with publication bias, earlier studies are likely to report larger SET/learning correlations. The second goal was to examine whether SET/learning correlations reported by authors with the most significant COI—authors working for or owning SET selling corporations—are larger than correlations reported by other authors. We expected that the corporate COIs would result in the largest SET/learning correlations. The third goal was to examine the size of the SET/learning correlations by the type of COI: corporate, administrative, evaluation unit, SET author and funder COIs. We made the following predictions as to which of these COIs would result in larger vs. smaller COI effects: (a) we expected the corporate COIs to result in the strongest effect, (b) we expected all COIs to have some effect, (c) we expected the SET author COI alone to have some but a weaker effect than the SET author COI combined with the corporate, administrative, or evaluation unit COI, (d) we expected the evaluation unit COI to be more significant that the SET author COI alone, and (e) we expected the administrative COI be similar to the evaluation unit COI but perhaps stronger given the administrations' need to evaluate faculty, and in particular, to do it efficiently, inexpensively, and with an aura of numerical objectivity. The fourth goal was to examine the associations between SET/learning correlations and authors' place

of employment/work including corporate, administrative, evaluation unit, education and psychology departments, business and economics departments, and other. With respect to authors' departments, our analyses were purely exploratory and aimed to examine *Clayson*'s (*2009*) suggestion that SET/learning correlations reported by authors from the education and liberal arts disciplines are larger than those reported by authors from business/economics departments. Our final goal was to explore if SET/learning correlations vary with the number of COIs that are present. Although COIs are unlikely to have similar or a strictly additive effect, we expected studies with a greater number of COIs to result in the larger SET/learning correlations. For example, as noted above, a SET author who established a corporation to sell his or her SET system is likely to have more substantial overall COI than a SET author who did not commercialize his or her SET system and had no other COI.

## METHOD

A preliminary review of all the available multisection studies revealed that the articles themselves rarely declared any COIs and often failed to even disclose sufficient information to determine whether any conflict of interest might exist. Accordingly, to determine the presence of any COIs, we adopted the method detailed in a recent investigation of COIs and their relationship to outcomes in randomized controlled trials (*Ahn et al., 2017*). We proceeded in three steps. First, we examined each article reporting a multisection study. Second, we examined all articles published by the study authors within five years of the multisection study publication date. Third, we searched Google for author's CVs, and other relevant publications.

For each multisection study reported in *Uttl, White & Gonzalez (2017)*, we coded for the presence or absence of any of the following direct and indirect COIs:

**Corporate interest.** If at least one author worked for a corporation or organization selling SET services (e.g., Educational Testing Service, IDEA), the corporate interest was coded as present.

**Evaluation unit interest.** If at least one author worked for a corporate or university teaching evaluation unit, the evaluation unit interest was coded as present.

**Administrative interest.** If at least one author was an administrator who was likely to be directly responsible for evaluation of faculty serving under the administrator, for example, a chair of the department or a dean, the administrative interest was coded as present.

**SET author interest.** If at least one author authored or co-authored a more widely used SET (i.e., SET used more widely than in one particular study), the SET author interest was coded as present.

**Funder interest**. If a funding organization had a direct COI (e.g., when SET corporation funded the study), the funder interest was considered present. Conversely, the funder interest was considered absent if a study was funded from granting agencies such as National Science Foundation or the Kellogg's Foundation that are not known to have any identifiable interest in the use of SET systems.

**Department type.** Each author's department was classified as one of the following: corporate unit, evaluation unit, education or psychology department, business or

economics department, and other/unknown depending on author affiliation stated in the publication itself.

Two independent coders coded COIs using the method described above and any disagreements were resolved by discussion and consensus. The agreement between the two coders was assessed by Cohen's kappa. Cohen's kappa ranges from −1 to 1 with 1 indicating perfect agreement, 0 chance agreement, and −1 perfect disagreement. In general, values 0.0 to 0.2 indicated slight agreement, 0.21 to 0.40 fair agreement, 0.41 to 0.60 moderate agreement, 0.61 to .80 substantial agreement, and 0.81 to 1 almost perfect agreement (*Hallgren, 2012*).

We examined the effects of publication period, COIs, and authors' department using the random effect model (using restricted maximum-likelihood estimator or REML and Fisher's z transformation of correlations) with specific moderators. All reported analyses were conducted using R (version 3.4.4) (*R Core Team, 2018*), and more specifically, using packages meta (version 4.9-2), metafor (version 2.0-0), metasens (0.3-2), and wPerm (version 1.0.1). One may argue that the assumption of bivariate normality required for our analyses is unlikely to have been satisfied. However, a number of studies have found that random effect meta-analysis estimates, including confidence interval estimates, are robust to even severe violations of the bivariate normality assumption (*Kontopantelis & Reeves, 2012*). Alternatively, our data could be analyzed using permutation tests. However, permutation tests are not suitable because they do not take into account differences in sample sizes of the primary studies. Nevertheless, our re-analyses of the data using permutation tests yielded similar results.

## RESULTS

Table 1 shows all 97 multisection studies included in *Uttl, White & Gonzalez*'s (*2017*) meta-analysis, with each study size ($n$) and SET/learning correlation for instructor ($r$) taken from Uttl et al.'s Table 2. The table includes a column showing the presence or absence of each conflict of interest: corporate (Corp), administrative (Admin), evaluation unit (Eval U.), SET author (SET Auth.), funder (Funder), and the total count of all COIs present for each study (i.e., the sum of corporate, administrative, evaluation unit, SET author, and funder interests). In addition, the table includes a column indicating whether or not authors were from education or psychology (E/Psy), or from business or economics departments (B/Econ). Inter-rater reliability measured by Cohen's kappa was nearly perfect; Cohen's kappa was 1.00 for corporate COI, 0.94 for administrative COI, 1.00 for evaluation unit COI, 0.98 for SET author COI, and 0.88 for funder COI. Any discrepancies between the two coders were discussed until a consensus was achieved and the consensus data are reported in Table 1.

Table 2 shows the means, *SD*s, and a matrix of the simple correlations among all variables including the COIs, department memberships, SET/learning correlation, study size, and publication period. The means indicate that one of the COIs—the funder COI—was rare, present in only 4% (4 out of 97) of the studies. The correlation matrix indicates that studies published in 1981 and after were associated with a lower prevalence of COIs and a higher

**Table 1   Studies included in *Uttl, White & Gonzalez*'s (*2017*) meta-analysis with identified COIs and authors' associations with education and psychology departments, and business and economics departments.**

| Study | n | r | Corp[1] | Admin[2] | Eval U.[3] | SET Auth.[4] | Funder[5] | No. of COIs[6] | E/Psy[7] | B/Econ[8] |
|---|---|---|---|---|---|---|---|---|---|---|
| Beleche.2012 | 82 | 0.21 | 0 | 0 | 0 | 0 | 0 | 0 | 0 | 1 |
| Benbassat.1981 | 15 | 0.18 | 0 | 0 | 0 | 0 | 0 | 0 | 0 | 0 |
| Bendig.1953a | 5 | 0.89 | 0 | 0 | 0 | 0 | 0 | 0 | 1 | 0 |
| Bendig.1953b | 5 | −0.80 | 0 | 0 | 0 | 0 | 0 | 0 | 1 | 0 |
| Benton.1976 | 31 | 0.17 | 0 | 0 | 0 | 1[S1] | 0 | 1 | 0 | 0 |
| Bolton.1979 | 10 | 0.68 | 0 | 0 | 0 | 0 | 0 | 0 | 1 | 0 |
| Braskamp.1979.01 | 19 | 0.17 | 0 | 1[A1] | 1[E1] | 0 | 0 | 2 | 0 | 0 |
| Braskamp.1979.02 | 17 | 0.48 | 0 | 1[A1] | 1[E1] | 0 | 0 | 2 | 0 | 0 |
| Bryson.1974 | 20 | 0.55 | 0 | 0 | 0 | 0 | 0 | 0 | 0 | 0 |
| Capozza.1973 | 8 | −0.94 | 0 | 0 | 0 | 0 | 0 | 0 | 0 | 1 |
| Centra.1977.01 | 7 | 0.60 | 1[C1] | 0 | 1[E2] | 1[S2] | 0 | 3 | 0 | 0 |
| Centra.1977.02 | 7 | 0.61 | 1[C1] | 0 | 1[E2] | 1[S2] | 0 | 3 | 0 | 0 |
| Centra.1977.03 | 22 | 0.64 | 1[C1] | 0 | 1[E2] | 1[S2] | 0 | 3 | 0 | 0 |
| Centra.1977.04 | 13 | 0.23 | 1[C1] | 0 | 1[E2] | 1[S2] | 0 | 3 | 0 | 0 |
| Centra.1977.05 | 8 | 0.87 | 1[C1] | 0 | 1[E2] | 1[S2] | 0 | 3 | 0 | 0 |
| Centra.1977.06 | 7 | 0.58 | 1[C1] | 0 | 1[E2] | 1[S2] | 0 | 3 | 0 | 0 |
| Centra.1977.07 | 8 | 0.41 | 1[C1] | 0 | 1[E2] | 1[S2] | 0 | 3 | 0 | 0 |
| Chase.1979.01 | 8 | 0.78 | 0 | 0 | 1[E3] | 0 | 0 | 1 | 0 | 0 |
| Chase.1979.02 | 6 | 0.19 | 0 | 0 | 1[E3] | 0 | 0 | 1 | 0 | 0 |
| Cohen.1970 | 25 | 0.42 | 0 | 0 | 0 | 1[S3] | 1[F1] | 2 | 0 | 0 |
| Costin.1978.01 | 25 | 0.52 | 0 | 1[A2] | 0 | 0 | 0 | 1 | 1 | 0 |
| Costin.1978.02 | 25 | 0.56 | 0 | 1[A2] | 0 | 0 | 0 | 1 | 1 | 0 |
| Costin.1978.03 | 21 | 0.46 | 0 | 1[A2] | 0 | 0 | 0 | 1 | 1 | 0 |
| Costin.1978.04 | 25 | 0.41 | 0 | 1[A2] | 0 | 0 | 0 | 1 | 1 | 0 |
| Doyle.1974 | 12 | 0.49 | 0 | 0 | 1[E4] | 1[S4] | 0 | 2 | 0 | 0 |
| Doyle.1978 | 10 | −0.04 | 0 | 0 | 1[E4] | 1[S4] | 0 | 2 | 0 | 0 |
| Drysdale.2010.01 | 11 | 0.09 | 0 | 0 | 0 | 0 | 0 | 0 | 1 | 0 |
| Drysdale.2010.02 | 10 | −0.02 | 0 | 0 | 0 | 0 | 0 | 0 | 1 | 0 |
| Drysdale.2010.03 | 8 | 0.64 | 0 | 0 | 0 | 0 | 0 | 0 | 1 | 0 |
| Drysdale.2010.04 | 11 | 0.03 | 0 | 0 | 0 | 0 | 0 | 0 | 1 | 0 |
| Drysdale.2010.05 | 10 | −0.23 | 0 | 0 | 0 | 0 | 0 | 0 | 1 | 0 |
| Drysdale.2010.06 | 12 | −0.1 | 0 | 0 | 0 | 0 | 0 | 0 | 1 | 0 |
| Drysdale.2010.07 | 11 | 0.19 | 0 | 0 | 0 | 0 | 0 | 0 | 1 | 0 |
| Drysdale.2010.08 | 16 | 0.41 | 0 | 0 | 0 | 0 | 0 | 0 | 1 | 0 |
| Drysdale.2010.09 | 11 | 0.23 | 0 | 0 | 0 | 0 | 0 | 0 | 1 | 0 |
| Elliot.1950 | 36 | 0.32 | 0 | 0 | 1[E5] | 0 | 0 | 1 | 0 | 0 |
| Ellis.1977 | 19 | 0.58 | 0 | 0 | 0 | 0 | 0 | 0 | 1 | 0 |
| Endo.1976 | 5 | −0.15 | 0 | 0 | 1[E6] | 0 | 0 | 1 | 0 | 0 |
| Fenderson.1997 | 29 | 0.09 | 0 | 0 | 0 | 0 | 0 | 0 | 0 | 0 |
| Frey.1973.01 | 8 | 0.91 | 1[C2] | 0 | 0 | 1[S5] | 0 | 2 | 0 | 0 |
| Frey.1973.02 | 5 | 0.6 | 1[C2] | 0 | 0 | 1[S5] | 0 | 2 | 0 | 0 |
| Frey.1975.01 | 9 | 0.81 | 1[C2] | 0 | 0 | 1[S5] | 1[F2] | 3 | 0 | 0 |

**Table 1** (*continued*)

| Study | n | r | Corp[1] | Admin[2] | Eval U.[3] | SET Auth.[4] | Funder[5] | No. of COIs[6] | E/Psy[7] | B/Econ[8] |
|---|---|---|---|---|---|---|---|---|---|---|
| Frey.1975.02 | 12 | 0.18 | 1[C2] | 0 | 0 | 1[S5] | 1[F2] | 3 | 0 | 0 |
| Frey.1975.03 | 5 | 0.74 | 1[C2] | 0 | 0 | 1[S5] | 1[F2] | 3 | 0 | 0 |
| Frey.1976 | 7 | 0.79 | 1[C2] | 0 | 0 | 1[S5] | 0 | 2 | 1 | 0 |
| Galbraith.2012a.01 | 8 | 0.23 | 0 | 0 | 0 | 0 | 0 | 0 | 0 | 1 |
| Galbraith.2012a.02 | 10 | 0.32 | 0 | 0 | 0 | 0 | 0 | 0 | 0 | 1 |
| Galbraith.2012a.03 | 12 | −0.07 | 0 | 0 | 0 | 0 | 0 | 0 | 0 | 1 |
| Galbraith.2012a.04 | 8 | 0.31 | 0 | 0 | 0 | 0 | 0 | 0 | 0 | 1 |
| Galbraith.2012a.05 | 8 | −0.13 | 0 | 0 | 0 | 0 | 0 | 0 | 0 | 1 |
| Galbraith.2012a.06 | 9 | −0.16 | 0 | 0 | 0 | 0 | 0 | 0 | 0 | 1 |
| Galbraith.2012a.07 | 13 | 0.11 | 0 | 0 | 0 | 0 | 0 | 0 | 0 | 1 |
| Galbraith.2012b | 5 | 0.29 | 0 | 0 | 0 | 0 | 0 | 0 | 0 | 1 |
| Greenwood.1976 | 36 | −0.11 | 0 | 0 | 0 | 1[S6] | 0 | 1 | 1 | 0 |
| Grush.1975 | 18 | 0.45 | 0 | 1[A3] | 0 | 0 | 0 | 1 | 0 | 0 |
| Hoffman.1978.03 | 75 | 0.29 | 0 | 0 | 0 | 0 | 0 | 0 | 1 | 0 |
| Koon.1995 | 36 | 0.3 | 0 | 1[A4] | 0 | 1[S7] | 0 | 2 | 1 | 0 |
| Marsh.1975 | 18 | 0.42 | 1[C3] | 0 | 1[E7] | 1[S8] | 0 | 3 | 0 | 0 |
| Marsh.1980 | 31 | 0.38 | 1[C3] | 0 | 1[E7] | 1[S8] | 0 | 3 | 0 | 0 |
| McKeachie.1971.01 | 34 | 0.06 | 0 | 1[A5] | 0 | 1[S9] | 0 | 2 | 1 | 0 |
| McKeachie.1971.02 | 32 | −0.20 | 0 | 1[A5] | 0 | 1[S9] | 0 | 2 | 1 | 0 |
| McKeachie.1971.03 | 6 | 0.10 | 0 | 1[A5] | 0 | 1[S9] | 0 | 2 | 1 | 0 |
| McKeachie.1971.04 | 16 | 0.25 | 0 | 1[A5] | 0 | 1[S9] | 0 | 2 | 1 | 0 |
| McKeachie.1971.05 | 18 | 0.55 | 0 | 1[A5] | 0 | 1[S9] | 0 | 2 | 1 | 0 |
| McKeachie.1978 | 6 | 0.20 | 0 | 0 | 1[E8] | 1[S9] | 0 | 2 | 0 | 0 |
| Mintzes.1977 | 25 | 0.38 | 0 | 0 | 0 | 0 | 0 | 0 | 0 | 0 |
| Morgan.1978 | 5 | 0.92 | 0 | 0 | 0 | 0 | 0 | 0 | 0 | 1 |
| Murdock.1969 | 6 | 0.77 | 0 | 0 | 0 | 0 | 0 | 0 | 1 | 0 |
| Orpen.1980 | 10 | 0.61 | 0 | 0 | 0 | 0 | 0 | 0 | 0 | 0 |
| Palmer.1978 | 14 | −0.17 | 0 | 0 | 0 | 0 | 0 | 0 | 0 | 1 |
| Prosser.1991 | 11 | −0.42 | 0 | 0 | 1[E9] | 0 | 0 | 1 | 0 | 0 |
| Rankin.1965 | 21 | −0.06 | 0 | 0 | 0 | 0 | 0 | 0 | 1 | 0 |
| Remmers.1949 | 53 | 0.28 | 0 | 0 | 1[E10] | 1[S10] | 0 | 2 | 0 | 0 |
| Rodin.1972 | 12 | −0.75 | 0 | 0 | 0 | 0 | 0 | 0 | 1 | 0 |
| Sheets.1995.01 | 58 | 0.15 | 0 | 0 | 0 | 0 | 0 | 0 | 0 | 1 |
| Sheets.1995.02 | 63 | −0.25 | 0 | 0 | 0 | 0 | 0 | 0 | 0 | 1 |
| Solomon.1964 | 24 | 0.30 | 0 | 0 | 1[E11] | 0 | 0 | 1 | 0 | 0 |
| Soper.1973 | 14 | −0.17 | 0 | 0 | 0 | 0 | 0 | 0 | 0 | 1 |
| Sullivan.1974.01 | 14 | 0.51 | 0 | 1[A6] | 0 | 0 | 0 | 1 | 0 | 0 |
| Sullivan.1974.04 | 9 | 0.57 | 0 | 1[A6] | 0 | 0 | 0 | 1 | 0 | 0 |
| Sullivan.1974.05 | 9 | 0.33 | 0 | 1[A6] | 0 | 0 | 0 | 1 | 0 | 0 |
| Sullivan.1974.06 | 16 | 0.34 | 0 | 1[A6] | 0 | 0 | 0 | 1 | 0 | 0 |
| Sullivan.1974.07 | 8 | 0.48 | 0 | 1[A6] | 0 | 0 | 0 | 1 | 0 | 0 |
| Sullivan.1974.08 | 6 | 0.55 | 0 | 1[A6] | 0 | 0 | 0 | 1 | 0 | 0 |
| Sullivan.1974.09 | 8 | 0.08 | 0 | 1[A6] | 0 | 0 | 0 | 1 | 0 | 0 |

**Table 1** (*continued*)

| Study | n | r | Corp[1] | Admin[2] | Eval U.[3] | SET Auth.[4] | Funder[5] | No. of COIs[6] | E/Psy[7] | B/Econ[8] |
|---|---|---|---|---|---|---|---|---|---|---|
| Sullivan.1974.10 | 14 | 0.42 | 0 | 1[A6] | 0 | 0 | 0 | 1 | 0 | 0 |
| Sullivan.1974.11 | 6 | −0.28 | 0 | 1[A6] | 0 | 0 | 0 | 1 | 0 | 0 |
| Sullivan.1974.12 | 40 | 0.40 | 0 | 1[A6] | 0 | 0 | 0 | 1 | 0 | 0 |
| Turner.1974.01 | 16 | −0.51 | 0 | 0 | 0 | 0 | 0 | 0 | 0 | 0 |
| Turner.1974.02 | 24 | −0.41 | 0 | 0 | 0 | 0 | 0 | 0 | 0 | 0 |
| Weinberg.2007.01 | 190 | 0.04 | 0 | 0 | 0 | 0 | 0 | 0 | 0 | 1 |
| Weinberg.2007.02 | 119 | −0.26 | 0 | 0 | 0 | 0 | 0 | 0 | 0 | 1 |
| Weinberg.2007.03 | 85 | −0.09 | 0 | 0 | 0 | 0 | 0 | 0 | 0 | 1 |
| Whitely.1979.01 | 5 | 0.80 | 0 | 0 | 1[E12] | 1[S11] | 0 | 2 | 0 | 0 |
| Whitely.1979.02 | 11 | −0.11 | 0 | 0 | 1[E12] | 1[S11] | 0 | 2 | 0 | 0 |
| Wiviott.1974 | 6 | −0.04 | 0 | 0 | 0 | 0 | 0 | 0 | 0 | 0 |
| Yunker.2003 | 46 | 0.19 | 0 | 1[A7] | 0 | 0 | 0 | 1 | 0 | 1 |

**Notes.**

[1] Corp = Corporate COI (0 = absent, 1 = present).
[2] Admin = Administrative COI (0 = absent, 1 = present).
[3] Eval U. = Evaluation Unit COI (0 = absent, 1 = present).
[4] SET Auth = SET Author COI (0 = absent, 1 = present).
[5] Funder = Funder COI (0 = absent, 1 = present).
[6] No. of COIs = Number of COIs.
[7] E/Psy = Author from Education or Psychology Departments (0 = No, 1 = Yes).
[8] B/Econ = Author from Business or Economics Departments (0 = No, 1 = Yes).
[C1] Centra worked for the Educational Testing Service (ETS) selling SIR SETs (*Centra, 1977*; *Centra, 2015*).
[C2] Frey founded Endeavor Information Systems Inc., the corporation selling the Endeavor Instructional Rating System (Endeavor) (*Endeavor Information Systems Inc., 1979*).
[C3] Marsh was President of Evaluation, Testing and Research Inc., the corporation selling Student Evaluation of Education instrument (SEE), a predecessor of SEEQ (*Marsh, Fleiner & Thomas, 1975*).
[A1] Head, Measurement and Research Division, University of Illinois (*Braskamp, Caulley & Costin, 1980*).
[A2] Director, Introductory Psychology Program, University of Illinois (*Bernstein, 2012*).
[A3] Director, Introductory Psychology Program, University of Illinois (*Bernstein, 2012*).
[A4] Administrative positions at University of Western Ontario (*Harry Murray, 2004*).
[A5] Chair, Department of Psychology, University of Michigan (*McKeachie, Lin & Mann, 1971*).
[A6] Dean, Junior Studies, Memorial University of Newfoundland (*Memorial University of Newfoundland, 1970*).
[A7] Chair, Department of Accountancy, Western Illinois University (*Yunker & Yunker, 2003*).
[E1] Measurement and Research Division, University of Illinois (*Braskamp, Caulley & Costin, 1980*).
[E2] Educational Testing Service (*Centra, 1977*; *Centra, 2015*).
[E3] Bureau of Evaluative Studies and Testing, Indiana University (*Chase & Keene, 1979*).
[E4] Measurement Services Center, University of Minnesota (*Doyle & Crichton, 1978*).
[E5] Division of Educational Reference, Purdue University (*Elliott, 1950*).
[E6] Bureau of Educational Research and Center to Improve Learning and Instruction (*Endo & Della-Piana, 1977*).
[E7] Evaluation, Testing and Research Inc. (*Marsh, Fleiner & Thomas, 1975*) and Evaluation Services, University of Southern California (*Marsh, 1980*).
[E8] Center for Research on Learning and Teaching, University of Michigan (*Mckeachie, Lin & Mendelson, 1979*).
[E9] Center for Learning and Teaching, University of Technology, Sydney (*Prosser & Trigwell, 1991*).
[E10] Division of Educational Reference, Purdue University (*Remmers, Martin & Elliott, 1949*).
[E11] Center for the Study of Liberal Education for Adults, Chicago (*Solomon, Rosenberg & Bezdek, 1964*).
[E12] Measurement Services Center, University of Minnesota (*Whitely & Doyle, 1979*).
[S1] Scott developed the Inventory of Student Perception of Instruction (ISPI) (*Benton & Scott, 1976*).
[S2] Centra developed Student Instructional Report (SIR) while working for ETS (*Centra, 2015*).
[S3] S. H. Cohen and Berger developed Student Instructional Rating Report (SIRR) (*Cohen & Berger, 1970*).
[S4] Doyle developed Student Opinion Survey (SOS) (*Doyle, 1972*; *Doyle & Crichton, 1978*; *Doyle & Whitely, 1974*).
[S5] Frey developed The Endeavor Instructional Rating System (*Endeavor Information Systems Inc., 1979*).
[S6] Greenwood co-developed the Student Evaluation of College Teaching Behaviors (SECTB) (*Greenwood et al., 1973*; *Greenwood et al., 1976*).
[S7] Murray developed Teacher Behaviors Inventory (TBI) (*Murray, 1983*).
[S8] Marsh developed Students' Evaluation of Educational Quality (SEEQ) (*Marsh, 1980*; *Marsh, Fleiner & Thomas, 1975*).
[S9] McKeachie co-developed Student Perception of Teaching and Teacher (SPTT) (*Isaacson et al., 1964*; *Isaacson et al., 1964*).
[S10] Remmers co-developed Purdue Rating Scale (PRS) (*Remmers, Martin & Elliott, 1949*).
[S11] Doyle developed Student Opinion Survey (SOS) (*Doyle, 1972*; *Whitely & Doyle, 1979*).
[F1] Funded by Education Developmental Program, Training Services, and Evaluation Services, Michigan State University (*Cohen & Berger, 1970*).
[F2] Funded in part by Endeavor Information Systems Inc. (*Frey, Leonard & Beatty, 1975*).

**Table 2 Means, standard deviations, and correlations among key variables.**

|  | M | SD | 1. | 2. | 3. | 4. | 5. | 6. | 7. | 8. | 9. | 10. |
|---|---|---|---|---|---|---|---|---|---|---|---|---|
| 1. n | 20.86 | 25.91 |  |  |  |  |  |  |  |  |  |  |
| 2. r | .24 | .38 | −.17 |  |  |  |  |  |  |  |  |  |
| 3. 1981+[1] | .29 | .46 | **.29** | **−.27** |  |  |  |  |  |  |  |  |
| 4. Corp[2] | .15 | .36 | −.16 | **.38** | **−.27** |  |  |  |  |  |  |  |
| 5. Admin[3] | .25 | .43 | −.03 | .12 | **−.26** | **−.25** |  |  |  |  |  |  |
| 6. Eval U.[4] | .24 | .43 | −.13 | .16 | **−.30** | **.36** | **−.21** |  |  |  |  |  |
| 7. SET Auth.[5] | .31 | .46 | −.11 | **.27** | **−.38** | **.64** | −.07 | **.41** |  |  |  |  |
| 8. Funder[6] | .04 | .20 | −.07 | .16 | −.13 | **.34** | −.12 | −.12 | **.31** |  |  |  |
| 9. COI count[7] | .99 | 1.07 | −.18 | **.39** | **−.51** | **.73** | .19 | **.60** | **.85** | **.34** |  |  |
| 10. E/Psy[8] | .30 | .46 | −.06 | −.03 | .08 | **−.22** | .15 | **−.36** | −.05 | −.14 | **−.21** |  |
| 11. B/Econ[9] | .20 | .40 | **.36** | **−.28** | **.55** | **−.21** | **−.22** | **−.28** | **−.33** | −.10 | **−.44** | **−.32** |

**Notes.**

Correlations with $p < .05$ are printed in bold.

[1] 1981+ = Study was published 1981 or later (0 = No, 1 = Yes).
[2] Corp = Corporate COI (0 = absent, 1 = present).
[3] Admin = Administrative COI (0 = absent, 1 = present).
[4] Eval U. = Evaluation Unit COI (0 = absent, 1 = present).
[5] SET Auth = SET Author COI (0 = absent, 1 = present).
[6] Funder = Funder COI (0 = absent, 1 = present).
[7] COI count = Number of COIs.
[8] E/Psy = Author from Education or Psychology Departments (0 = No, 1 = Yes).
[9] B/Econ = Author from Business or Economics Departments (0 = No, 1 = Yes).

involvement of business and economics departments. Importantly, the correlation matrix also indicates that some COIs frequently co-occurred with other COIs. More detailed analyses revealed that out of 30 studies with the SET author COI, 15 studies included the corporate COI, and out of the remaining 15 studies with the SET author COI, 6 studies included the administrative COI, 6 studies included the evaluation unit COI, 1 study included the funder COI, and only 2 studies included no other COI. Most critically, the size of SET/learning correlations was associated with publication period (studies published in 1981 and after reported lower SET/learning correlations), and with corporate COIs, SET author COIs, and COI count.

Figure 1A, shows the magnitude of SET/learning correlations as a function of multisection study size (*Uttl, White & Gonzalez, 2017*). This figure shows that (1) the number of sections included in multisection studies was generally small with many studies based on as few as five sections, (2) many studies reported impossibly high correlations, and (3) the magnitude of SET/learning correlations decreased for larger sized studies (*Uttl, White & Gonzalez, 2017*). Figure 1B, shows the histogram of multisection studies by publication year. A large number of studies was published just prior to *Cohen*'s (*1981*) meta-analysis and a relatively smaller number of studies have been published since then. Figure 1C, shows the number of studies published prior to 1981 and since then by origin of the study. Prior to 1981, most studies were published by authors from SET corporations, evaluation units, administration, and education and psychology departments. In contrast, since 1981, most studies were published by authors from business and economics departments and psychology departments. No studies were published by

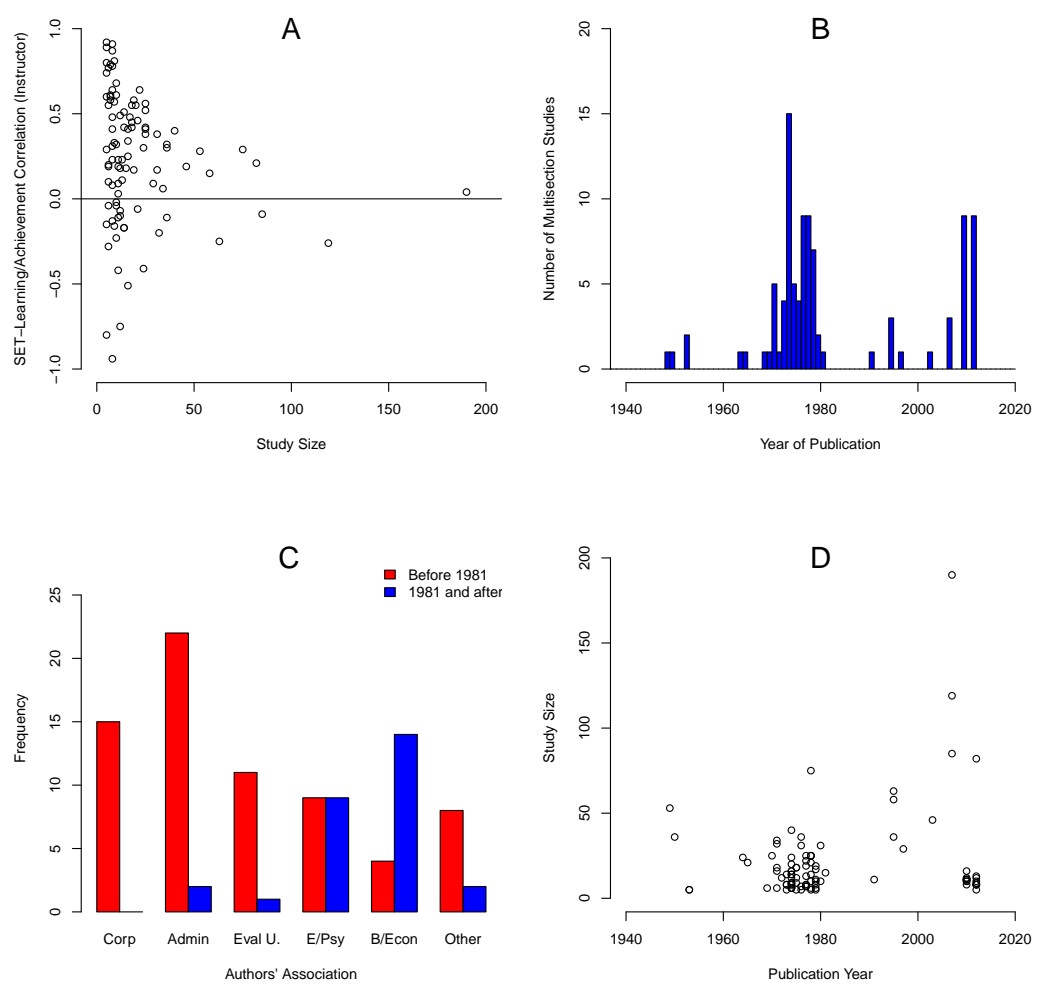

**Figure 1** **Descriptive analyses of SET/learning correlations, sample size, and publication year.** (A) The magnitude of SET/learning correlations as a function of the multisection study size (see *Uttl, White & Gonzalez, 2017*). (B) The histogram of multisection studies by year of publication. (C) The number of studies published prior to 1981 and since then by origin of the study. (D) Study size as a function of year of publication.

authors from SET corporations or from evaluation units. Figure 1D, shows the study size as a function of publication year. SET/learning correlations reported by studies published prior to 1981 vs. since then used smaller samples. That is, most studies published prior to 1981 used very small samples and larger samples became more common only after 1981.

Figure 2A, shows the size of SET/learning correlations as a function of publication date, prior to 1981 vs. 1981 and after. Each dot represents one SET/learning correlation superimposed on the boxplot of their distribution. Studies published prior to 1981 reported much larger SET/learning correlations than studies published in 1981 and after. The random effect meta-analysis (using z-transformed correlations and REML estimation method) with publication period as a moderator showed significant differences between SET/learning correlations reported by studies published prior to 1981 and studies published 1981 and

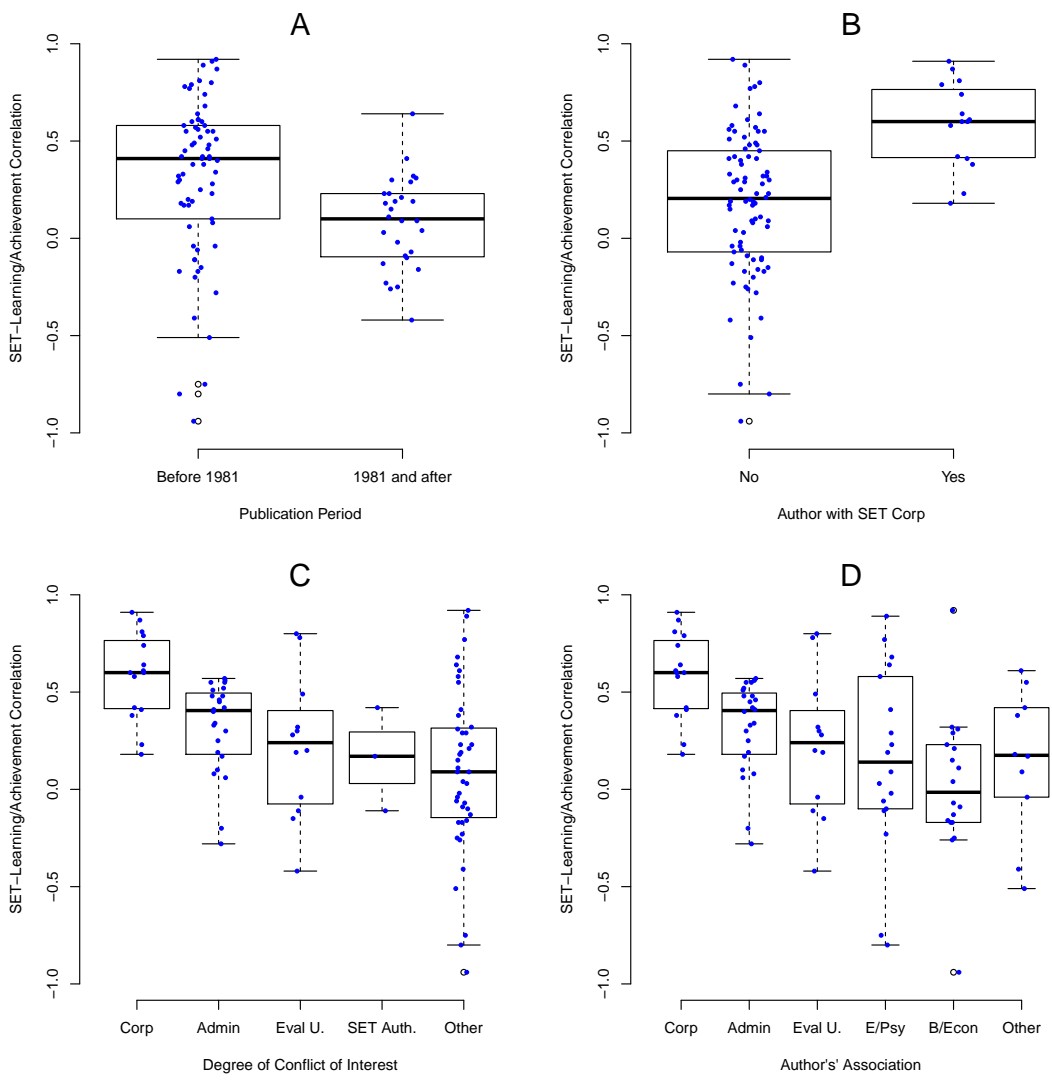

**Figure 2  SET/learning correlations as a function of publication time, COIs, and authors' associations.**
(A) The size of SET/learning correlations as function of publication date, prior to 1981 vs. 1981 and after. Each dot represents one SET/learning correlation superimposed on the boxplot of their distribution. (B) Boxplots of SET/learning correlations by authors' associations with SET corporations. (C) Boxplots of SET/learning correlations by authors' associations with SET corporations, administration, evaluation units, and authoring SET. For these analyses, studies were assigned in categories by the highest, most significant COI in the following order: corporate, administrative, evaluation unit, SET author, and no identified COI/other. (D) Boxplots of SET/learning correlations by authors' associations with SET corporations, administration, evaluation units, psychology/education departments, business/economics departments, and other departments. Similarly to the previous analysis, the highest level association was used to categorize studies.

after, $Q(1) = 10.31$, $p = .0013$. For the 69 studies published prior to 1981, $r = .31$, 95% CI [.22–.39], $I^2 = 53.1\%$ whereas for 28 studies published in 1981 or after, $r = .06$, 95% CI [−.07–.19], $I^2 = 19.2\%$.

Figure 2B, shows boxplots of SET/learning correlations by authors' associations with SET corporations. SET/learning correlations were much larger when at least one author was associated with a SET corporation. The random effect meta-analysis (using z-transformed correlations and REML estimation method) with corporate COI as a moderator showed significant differences between SET/learning correlations reported by studies with vs. without corporate COI, $Q(1) = 15.38$, $p < .0001$. For the 15 studies with corporate COI, the correlation was $r = .58$, 95% CI [.41–.71], $I^2 = 5.8\%$, whereas for the 82 studies without corporate COI, the correlation was $r = .18$, 95% CI [.10–.25], $I^2 = 52.7\%$.

Figure 2C, shows boxplots of SET/learning correlations by authors' associations with SET corporations, administration, evaluation units, and authoring SET. For these analyses, studies were assigned to categories by the highest, most significant COI present in the following order: corporate, administrative, evaluation unit, SET author, and no identified COI. This ordering was determined based on (a) our a priori expectations detailed in the introduction, (b) the findings that COIs frequently co-occurred, as detailed above, and rendered the comparative analyses of single COI effects impossible, (c) the finding that the funder COI was rarely present and when present it typically co-occurred with the corporate COI, and (d) our guess that the administrative COI might be stronger than the evaluation unit COI given that administrative interests are frequently cited as one of the key reasons for the widespread use of the SET. As the panel shows, the corporate COI was associated with the largest effects, the administrative COI with the next largest effects, the evaluation unit COI with smaller effects, the SET author COI with a still smaller effects, and the smallest effects were reported by studies with no known COI. The random effect meta-analysis with COI degree as a moderator showed significant differences between the groups of studies, $Q(4) = 28.54$, $p < .0001$. Table 3 highlights that corporate COI resulted in the largest effect of $r = .57$, administrative COI in $r = .33$, evaluation unit COI in $r = .25$, SET author COI in $r = .15$, and no known COI in $r = .06$. Neither the SET author COI (alone) nor the 'no known COI' estimates significantly differed from zero. We have also examined the association between SET/learning correlations and the specific ordering of COI categories above using a permutation relationship test (using R package wPerm), with 9,999 replications and Spearman's rank correlation method. Correlation between SET/learning correlations and the ordering of COI categories was moderately strong, $r = .45$, $p < .001$. Because we did not specify the relative order of the administrative and the evaluation unit COI categories a priori, we repeated the permutation test with the order of these two COI categories reversed. Correlation was nearly identical, $r = .41$, $p < .001$.

Figure 2D, shows boxplots of SET/learning correlations by authors' associations with SET corporations, administration, evaluation units, psychology/education departments, business/economics departments, and other departments. Similarly to the previous analysis, the highest level of association was used to categorize studies. The figure highlights that authors associated with education and psychology, and with business and economics, reported similar SET/learning correlations when authors involved in administration and in evaluation units were classified in administration and evaluation units rather than as ordinary members of their departments. The random effect meta-analysis with authors'

Table 3 Random effect meta-analysis of SET/learning correlations by authors' associations with the most significant COI: Subgroup results.

| Authors' association | $k$ | $r$ | 95% CI | $Q$ | $I^2$ |
|---|---|---|---|---|---|
| Corporate | 15 | .57 | [.41, .70] | 14.85 | 5.8% |
| Administrative | 24 | .33 | [.21, .45] | 21.76 | 0% |
| Evaluation unit | 12 | .25 | [.03, .44] | 11.53 | 4.6% |
| SET author | 3 | .15 | [−.16, .43] | 4.18 | 52.1% |
| None | 43 | .06 | [−.04, .16] | 105.30 | 60.1% |

Table 4 Random effect meta-analysis of SET/learning correlations by authors' associations with the most significant unit/department: Subgroup results.

| Authors' association/COI | $k$ | $r$ | 95% CI | $Q$ | $I^2$ |
|---|---|---|---|---|---|
| Corporate | 15 | .57 | [.41, .70] | 14.85 | 5.8% |
| Administrative | 24 | .33 | [.21, .45] | 21.76 | 0% |
| Evaluation unit | 12 | .25 | [.04, .44] | 11.53 | 4.6% |
| Education/Psychology | 18 | .15 | [−.02, .31] | 36.88 | 53.9% |
| Business/Economics | 18 | −.04 | [−.18, .10] | 39.88 | 57.4% |
| Other | 10 | .16 | [−.04, .34] | 23.26 | 61.3% |

association as a moderator showed significant differences between the groups of studies, $Q(5) = 33.82$, $p < .0001$. Table 4 highlights that authors associated with education and psychology reported $r = .15$, authors associated with business and economics reported $r = −.04$, and authors with no known associations reported $r = .16$.

Figure 3 shows boxplots of SET/learning correlations by the total number of COIs identified for each study. The figure highlights that studies with no COIs reported on average nearly zero SET/learning correlations whereas studies with 1, 2, or 3 COIs reported on average small to moderately large SET/learning correlations. The random effect meta-analysis with the number of COIs as a moderator showed significant differences between the groups of studies, $Q(3) = 22.35$, $p < .0001$. Table 5 highlights that studies with no identifiable COI resulted in estimated $r = .06$ whereas studies with the most COIs resulted in estimated $r = .53$. We have examined an association between SET/learning correlations and a number of COIs using permutation relationship test (using R package wPerm), with 9,999 replications and Spearman's rank correlation method. Correlation between SET/learning correlations and a number of COIs was moderately strong, $r = .39$, $p < .001$.

## DISCUSSION

Researchers with a COI found, on average, large positive SET/learning correlations. In contrast, researchers with no identifiable COIs found that SET/learning correlations were zero or nearly zero. As predicted, the largest SET/learning correlations were reported by authors with ties to corporations that sell SET systems. Smaller but still substantial SET/learning correlations were reported by researchers with administrative assignments and with evaluation units/departments responsible for the administration of SET. Moreover, authors with the most significant COIs were publishing their studies primarily prior to
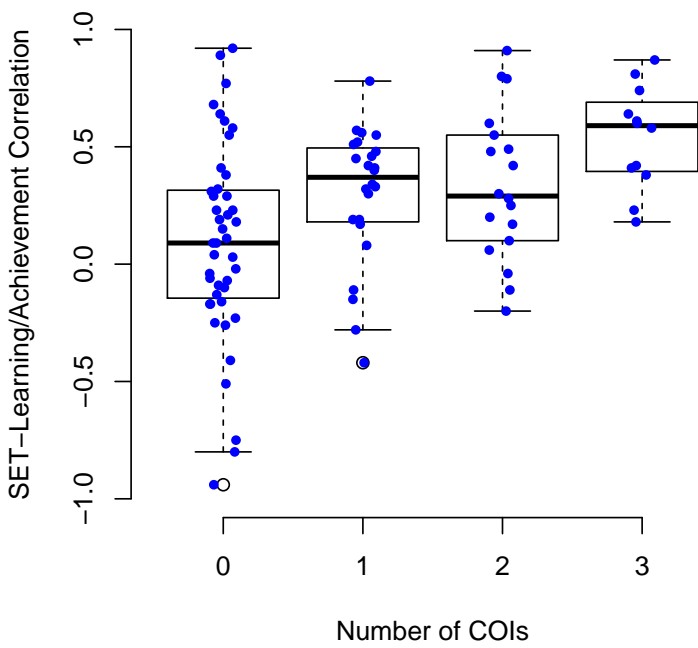

**Figure 3** **SET/learning correlations by the total number of COIs identified for each study.** The figure shows SET/learning correlations as a function of number of COIs. Each dot represents one SET/learning correlation superimposed on the boxplot of their distribution.

**Table 5** **Random effect meta-analysis of SET/learning correlations by number of COIs present: sub-group results.**

| Number of COIs | $k$ | $r$ | 95% CI | $Q$ | $I^2$ |
|---|---|---|---|---|---|
| 0 | 43 | .06 | [−.05, .16] | 105.30 | 60.1% |
| 1 | 24 | .33 | [.20, .45] | 23.88 | 3.7% |
| 2 | 18 | .30 | [.14, .44] | 26.25 | 35.2% |
| 3 | 12 | .53 | [.34, .67] | 9.50 | 0% |

1981 whereas authors with no or less significant COIs were publishing their studies in 1981 or afterwards. Studies published prior to 1981 reported small but significant ($r = .31$) SET/learning correlations whereas studies published in 1981 and after reported near zero, non-significant SET/learning correlations ($r = .06$). As our analyses show, the presence of COIs was associated with earlier publication date but also with smaller samples. Finally, whereas corporate, administrative, and evaluation unit authors nearly ceased publishing multisection studies on SET/learning correlations, authors from business and economics departments are now responsible for a substantial portion of newer, larger, and higher quality studies published since 1981.

Our findings are striking but at the same time not surprising. As *Ahn et al. (2017)* recently showed, even effects reported in randomized clinical trials—a gold standard of experimental design used to determine the effectiveness of therapies—are correlated with authors' conflict of interest. When a principal investigator has financial ties with a

drug manufacturer, a study was more than three times as likely to report positive drug effects than when the principal investigator had no such financial ties. Just like drug manufacturers, SET corporations need to sell SET systems to as many universities and colleges as they can, as their profits as well as their presidents', scientists', and employees' salaries depend on such sales. SET corporations reporting that SET/learning correlations are low or near zero would terminate their SET related revenues and reasons for their existence. Similarly, evaluation units need to show their value to the administration and reporting that SET ratings do not correlate with learning may result in the demise of these units. Moreover, as argued previously, administrators need inexpensive and efficient ways to evaluate faculty and if administrators report that SET do not correlate with learning they could not justify the use of such ratings. In turn, they would have to search for more costly and more demanding alternative ways to evaluate faculty's teaching effectiveness. To start, they would have to figure out what effective teaching is, something that educators were unable to establish despite decades of sustained efforts, and how to measure effective teaching both reliably and with a high degree of validity necessary for making high stakes personnel decisions (*Uttl, White & Gonzalez, 2017*). This definition and measurement problem has been brought to the forefront by a recent arbitration decision in *Ryerson University v. Ryerson Faculty Association* (2018 CanLII 58446) where an arbitrator ordered Ryerson University "to ensure that FCS [Ryerson's acronym for SET] results are not used to measure teaching effectiveness in promotion or tenure."

Finally, the COI influence on SET research findings is likely facilitated by a general lack of transparency and openness evidenced in this literature. SETs are typically not made public and only SET corporations, evaluation units, and administrators have access to SET data. As a result, the findings originating from the SET corporations, evaluation units, and administrators cannot be examined and verified by independent researchers. Given that most authors did not disclose COIs and their COIs had to be determined by searching other publicly available sources, it is likely that we missed some COI ties and, in turn, this may have underestimated the strength of associations between COIs and the reported SET/learning correlations. In future studies, it is desirable that authors declare COIs clearly and with specificity within each study. Importantly, requirements for authors to declare COIs have become more strict and explicit over the years and this ought to facilitate the detection of COIs. Similarly, requirements for transparency have also became more prominent, with some journals (e.g., *PLOS*, *PeerJ*) requiring data to be publicly posted prior to an article publication, thus providing instantaneous opportunities for other researchers to examine reported analyses and findings.

Our findings suggest that COIs, including corporate and administrative interests, had a major influence on findings of multisection studies, especially the findings of multisection studies published prior to 1981, prior to *Cohen*'s (*1981*) meta-analysis that concluded that SET/learning correlations were substantial. When *Cohen (1981)* worked on his meta-analysis of SET/learning correlations in multisection studies, he did not notice apparent COIs in many of these studies, and as we noted above, he himself had an interest in demonstrating the validity of SET/learning correlations as a measure of teaching effectiveness. P. A. Cohen's COI may explain why he did not notice strong evidence of

small sample bias (*Uttl, White & Gonzalez, 2017*) and why he disregarded dependence of SET/learning correlations on sample size even though reviewers of his meta-analysis were "concerned that rating/achievement correlations vary according to the number of sections used in the study" (*Cohen, 1981*, p. 303).

## CONCLUSIONS

SET/learning correlations reported by multisection studies vary substantially with COI ties of the authors of these studies. Authors with ties to SET selling corporations, administration, and evaluation units found small to moderately large SET/learning correlations. In contrast, authors with no identifiable COIs found zero or near zero SET/learning correlations. The large variability in findings associated with COIs is troubling. Thousands of faculty members have been evaluated using SETs and many have had their careers terminated by SETs. Yet, the evidence is now clear that SETs do not measure teaching effectiveness (*Uttl, White & Gonzalez, 2017*), are influenced by a number of teaching effectiveness irrelevant factors (TEIFs) not attributable to professors (e.g., student interest, student prior knowledge, subject matter, class size), and are also influenced by a number of faculty attributes—sex, accent, national origin, beauty/hotness—that universities are ill-advised to use in high stakes personnel decisions because use of such attributes is at minimum illegal.

Our findings highlight the need for openness and transparency, especially when research is likely to be used to support high stakes personnel decisions. At minimum, the COI ought to be clearly declared, including ownerships of shares in SET selling corporations and salaries derived from such activities, and data made available for other researchers wishing to verify reported findings. Although one may hope that *Cohen*'s (*1981*) meta-analysis would not be published today without declaration of COI and without Cohen providing at minimum a list of all studies included in his meta-analysis, the SET/learning correlations he extracted for each study, and the number of sections each SET/learning correlation was based on, this is not guaranteed.

Finally, we need to encourage authors with no COI ties to conduct relevant research. As noted above, many newer multisection studies were conducted by authors from business and economics departments and by other authors with no COI ties. Without their contributions we may still erroneously believe that SET scores are valid measures of faculty's teaching effectiveness.

### Funding
The authors received no funding for this work.

### Competing Interests
The authors declare there are no competing interests.

## Author Contributions

- Bob Uttl conceived and designed the experiments, performed the experiments, analyzed the data, contributed reagents/materials/analysis tools, prepared figures and/or tables, authored or reviewed drafts of the paper, approved the final draft.
- Kelsey Cnudde and Carmela A. White performed the experiments, analyzed the data, contributed reagents/materials/analysis tools, prepared figures and/or tables, authored or reviewed drafts of the paper, approved the final draft.

## Data Availability

The raw data are available in Table 1 and as a Supplemental File.

## Supplemental Information

Supplemental information for this article can be found online at http://dx.doi.org/10.7717/peerj.7225#supplemental-information.

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
