# Peer review of "Conflict of interest explains the size of student evaluation of teaching and learning correlations in multisection studies: a meta-analysis"

_PeerJ, doi:10.7717/peerj.7225_

## Round 0.1 · original submission · Major Revisions

Hi Bob and colleagues,

Thank you for your submission to PeerJ.

My apologies for making you wait this long for a decision on your submission. The delay is entirely my fault, and in no way to be blamed on PeerJ or the reviewers.

It is my opinion as the Academic Editor for your article - Conflict of interest explains the size of student evaluation of teaching and learning correlations in multisection studies: A meta-analysis - that it requires Major Revisions.

I will briefly outline what regard as the most crucial issues that need to be addressed in a revision.

• First, please delete from the manuscript the discussion of the IDEA Center’s reaction to the Uttl et al. criticism of SET. This discussion is out of place in the present manuscript. Moreover, this discussion detracts from the focus on how SET may be influenced by conflicts of interests.

• Second, the manuscript requires a clear discussion and definition of what you mean by conflict of interest (COI). CsOI come in many different forms (e.g., financial, legal) and can be real or perceived. If not all conflicts are comparable in some manner, is it still proper to count them like apples, as was done in Table 1?

• Third, the manuscript needs to include a clear statement about how COI was operationalized for the purpose of your study. Both reviewers have asked for more specific, objective information about how a decision about the presence or absence of CsOI was made in each case. This type of information needs to be included either in the manuscript or in an accessible supplement to the manuscript. Without (and even with) access to such information, there is likely to be substantial disagreement about the presence/absence of real or apparent conflicts in some studies, for example, those authored or co-authored by individuals in administrative positions.

• Fourth, the manuscript needs to be more explicit about the ordering of COI severity. Reviewer 2 suggests statistical tests that may be used to strengthen the COI ordering, if more was intended than an after-the-fact ordering based on the strength of associations.

• Fifth, Reviewer 2 draws your attention to additional multi-section studies that ought to have been included in your review of the literature.

• Sixth, Reviewer 1 draws attention to the fact that there is much more recognition of the need for disclosing CsOI today than 10, 20 or 30 years ago. This change in reporting practice ought to be acknowledged and discussed in the manuscript.

I have designated this submission as requiring Major Revisions because of the work that is likely to be required for responding to the above listed points, especially Points 2 and 3. I have also chosen Major Revisions because this manuscript, in its present form, is an opinion piece which raises a host of complex ethical and legal issues (and in addition, opinion pieces are out of scope at PeerJ). I am fairly certain that many or all of these issues may be managed, however, by a careful revision which acknowledges the changed practices on disclosing COIs, and which makes explicit the objective criteria which were used for determining the presence/absence of each type of COI in each publication .

·

Basic reporting

The coverage is good, but the paper omits a pivotal small multi-section study published in 2014 by MacNell et al. and a large multi-section study published in 2016 by Boring et al. (also reported by Boring in 2017: https://www.sciencedirect.com/science/article/abs/pii/S0047272716301591). Those studies are generally consistent with the findings of the current ms., though: negative association, and the authors were from Anthropology, Sociology, Economics, and Statistics. However, the methodology in the current paper would have counted the Boring et al. (2016) study as 2 COI and the Boring (2017) study as one COI (Boring and Stark held administrative positions as well as academic positions; Boring was responsible for SET at her institution). I think there were no COIs for the MacNell et al. study.

Experimental design

Non-experimental. Research questions were well defined. It was not clear whether the ordering of the severity of COI was developed a priori (before computing the correlations) or was simply reporting the order of the strengths of association. In particular, it was surprising to me that SET instrument authorship was less strongly associated with the correlation than was role as an administrator, if the COI hypothesis is correct (since it seems like a stronger COI).

If the ranking is intended to be a conclusion, a permutation test of the association between "strength of COI" and "strength of correlation" could be performed. Within a parametric framework, one could assess the confidence that the empirical ranking is "correct."

I have not used the R packages the paper mentions, so I do not happen to know whether there are parameter settings one needs to select, nor whether the authors used default settings. The paper should give the versions of all software packages and libraries used (R, meta, metafor, and metasens).

Other than that, the analysis seemed replicable from the tables.

Supporting data to show, paper by paper, the reasons the authors arrived at the particular COIs (columns 4--10 of Table 1) would be valuable, as would making Table 1 available in machine-readable electronic form (e.g., .csv).

Validity of the findings

The papers within the meta-analysis seem to be an appropriate collection to study, if not exhaustive.

The reliance on meta-analysis (with or without random effects) as a statistical technique does not make the paper more persuasive, because the assumptions of meta-analysis are baroque and disconnected from reality.

I think the correlations and figures tell the story on their own. If the authors want to assess some measure of statistical significance of the associations, a suitable permutation test would be more appropriate and convincing than meta-analysis with random effects. In particular, it could test hypotheses such as "year of publication is irrelevant to the correlation the study finds" or "number of COIs is irrelevant to the correlation the study finds," etc., without requiring the assumption that different studies represent independent samples from normal distributions of normalized effect sizes, with means that (may) depend on other covariates.

The tables need more informative captions, including explanations of the variables.

Additional comments

Boring et al. (2016) has two online reviews, one positive and one neutral. The neutral review is from an employee of IDEA, Jason Barr. IDEA also blogged about the Boring et al. paper, also misattributing quotations from the literature to the authors: https://www.ideaedu.org/Portals/0/Uploads/Documents/Response_to_Bias_Against_Female_Instructors.pdf

The IDEA blog author evidently misread Boring et al., since most of the criticisms in the blog post are contradicted by the paper itself.

Hilariously, the IDEA blog criticized the fact that the Boring et al. paper was post-reviewed, despite (1) the fact that their own employee was one of the post reviewers, and (2) many of IDEA's papers used as evidence that there is a positive association between SET and learning are self-published tech reports that were not peer reviewed at all.

·

Basic reporting

.

Experimental design

.

Validity of the findings

.

Additional comments

Attached my review of the article by Uttl et al.

---

## Round 0.2 · Minor Revisions

Hi Bob,

Thanks for the revision of your submission, and for your responsiveness to the reviewers' comments.

Your revisions was sent back to the original reviewers, and both of them have made additional valid comments that required your attention and consideration.

I begin by drawing your attention to comments by Reviewer 1. Reviewer 1 notes that your correlation confidence intervals are not meaningful because of mistaken assumptions about distributions. I recommend that you conduct the permutation tests recommended by the reviewer. Alternatively, you might know of evidence which shows that the confidence intervals you provide are valid even if distributional assumptions are violated.

Reviewer 1 also notes that the permutation tests used for the correlation between COI and strength of associations are not valid. I have no doubt that you have the skill for carrying out the additional work recommended.

Finally, Reviewer 1 comments on the quality of the data, on the information sources used for identifying CsOI. The revision goes a long way toward clarifying the method used for identifying conflicts of various kinds. However, questions remain and more information is required to make transparent exactly how each kind of conflict was identified. In some cases, it would seem that you needed access to authors' CV, or that you needed to contact authors to clarify their former corporate associations or administrative roles and responsibilities (at the time of publication of the research). Also, is it possible to estimate missed conflicts because of a lack of access to relevant records?

Because of such concerns about the quality of the COI classification, Reviewer 1 recommends a coarser taxonomy which would likely avoid most of the concerns raised in the review and above. I urge you to consider this option especially if it is not possible to be completely transparent about how conflicts were established in a valid and reliable manner.

In keeping with this issue, in a final revision of the manuscript, I would like to see a statement about the reliability of the method used for identifying CsOI.

Reviewer 2 also makes a number of valid and helpful recommendations. The most important of these concerns the 2 quotations at the beginning of the manuscript. For the reasons given by the reviewer, I agree that those quotations ought to be removed. I have no doubt that in their absence, readers will be more sensitive to the broader range of factors, not just monetary influences, that can contribute to conflicts.

Reviewer 2 also recommends that you streamline the manuscript by removing figures and tables which are not absolutely necessary. I agree with this recommendation. Too much information is likely to detract from the core message which is emphasized in the discussion.

Although neither reviewer remarked on it, I also recommend that you delete your label 'voodoo' for high correlations. The effect of such a label is difficult to predict, but it definitely does not clarify your message or the size of the correlations.

Finally, the manuscript needs careful copy editing.

No doubt you are not thrilled to read this call for additional revisions. However, because your manuscript focuses on an issue which is highly sensitive and profoundly important for making decisions about careers, promotions, remuneration, teaching practices, etc., all aspects of it need to be solid and based on the best available methods.

·

Basic reporting

No comment.

Experimental design

# I still cannot endorse the statistical analyses:

1. The confidence intervals for correlation coefficients are based on the assumption that the data are samples from bivariate normal distributions, which is false. The confidence intervals are meaningless. Permutation tests of the hypothesis that the correlation is zero would be more appropriate for this part of the analysis.

2. The permutation tests for correlation between COI and strength of association are not valid because the COI scale was established post hoc, after examining the data. This could be accounted for by maximizing the correlation coefficient over partial re-orderings of the COI scale, in each iteration of the simulation. That will require some coding, but not a great deal.

# There is also a real question about data quality. Author affiliations generally do not include enough information to determine whether some of the COIs considered in the manuscript exist. For instance, in Boring et al. (2015), Boring had a research position but also had institutional responsibility for SET at SciencesPo, and Stark was department chair, responsible for evaluating faculty for employment decisions. Neither institutional role would be evident from looking at the paper. Boring's role would not be ascertainable from a web search, and a current web search would not uncover Stark's prior role. I have no reason to believe that there is no similar issue with the publications analyzed in the present manuscript. I think a cruder taxonomy would be easier to defend (works for a company that sells SET, designed an SET instrument, both, or neither). Within universities, faculty administrative roles are ephemeral and hard to ascertain.

Validity of the findings

I think the overall qualitative conclusions are true, but the quantitative conclusions are not valid: they are based on misapplications of statistical methods. They ignore distributional assumptions and do not take into account the post facto tuning of the COI scale.

Additional comments

I really like the hypothesis and the general approach--and I believe the conclusions--but I can't overlook the misuses of statistical methods. I'm truly sorry.

·

Basic reporting

The manuscript is very readable and the authors responded well to criticism of their earlier version. I have only two comments. I think that the two money quotes at the beginning of the paper are inappropriate. First they would only apply to publications by SET publishers and not the other COIs, second they imply that COI resulted in deliberate biasing of findings. The ample literature about the influence of corporate funding on scientific results has never shown (nor assumed) that the bias was due to deliberate decisions by researchers. And the authors are careful in not suggesting this either. Therefore, I suggest that they skip the two quotations at the start of the paper.


I also wondered whether the authors did not engage in a bit of overkill in presenting their results in figures and tables. For example, in Figure 2, panels C and D appeared to me a bit redundant. I think that the bottom right panel could be eliminated without great loss of information. Similarly, the random effect meta-analyses are sufficiently discussed in the text as to make Tables 3 and 4 redundant.

Experimental design

not relevant

Validity of the findings

The findings are extremely relevant

Additional comments

The Uttl et al. (2017) meta analysis was an extremely important publication. Although there was overwhelming evidence that SETs were biased, the fact that they seemed to correlate with grades in between class comparisons suggested that there was some degree of validity, which probably could be uncovered, if one eliminated the biasing factors statistically (i.e., Greenwald & Gillmore, 1997). After their meta-analysis, this procedure seems unlikely to be successful. The present paper is not only and important addition to the literature on the (lack of) validity of the SET, it also adds to the growing literature on the influence of COI on research findings. Thus, I hope that it will be published soon so that I can cite it.

---

## Round 0.3 · Minor Revisions

Hi Bob,
I am good with the way you have addressed the reviewers and my comments in the most recent revision. However, the article still will profit from final copy editing. With this in mind, I have gone through the article and recommended revisions, all tracked for you. I urge you to review all of these minor edits and then submit the final to the journal. There is not need for me to see it again.

---

## Round 0.4 · accepted · Accept

This is an important contribution, and I look forward to seeing it in print. Congratulations, and thank you for going along with my repeated requests for revisions.